# The Most Competent Plant-Derived Natural Products for Targeting Apoptosis in Cancer Therapy

**DOI:** 10.3390/biom11040534

**Published:** 2021-04-03

**Authors:** Sadegh Rajabi, Marc Maresca, Alexei Valerievich Yumashev, Rasool Choopani, Homa Hajimehdipoor

**Affiliations:** 1Traditional Medicine and Materia Medica Research Center, Shahid Beheshti University of Medical Sciences, Tehran 1434875451, Iran; 2Aix Marseille Univ, CNRS, Centrale Marseille, iSm2, 13397 Marseille, France; 3Department of Prosthetic Dentistry, Sechenov First Moscow State Medical University, 119991 Moscow, Russia; sfmsmu@mail.ru; 4Department of Traditional Medicine, School of Traditional Medicine, Shahid Beheshti University of Medical Sciences, Tehran 1516745811, Iran; 5Traditional Medicine and Materia Medica Research Center and Department of Traditional Pharmacy, School of Traditional Medicine, Shahid Beheshti University of Medical Sciences, Tehran 1516745811, Iran

**Keywords:** plant extracts, phytochemical, medicinal herb, apoptosis, cancer

## Abstract

Cancer is a challenging problem for the global health community, and its increasing burden necessitates seeking novel and alternative therapies. Most cancers share six basic characteristics known as “cancer hallmarks”, including uncontrolled proliferation, refractoriness to proliferation blockers, escaping apoptosis, unlimited proliferation, enhanced angiogenesis, and metastatic spread. Apoptosis, as one of the best-known programmed cell death processes, is generally promoted through two signaling pathways, including the intrinsic and extrinsic cascades. These pathways comprise several components that their alterations can render an apoptosis-resistance phenotype to the cell. Therefore, targeting more than one molecule in apoptotic pathways can be a novel and efficient approach for both identifying new anticancer therapeutics and preventing resistance to therapy. The main purpose of this review is to summarize data showing that various plant extracts and plant-derived molecules can activate both intrinsic and extrinsic apoptosis pathways in human cancer cells, making them attractive candidates in cancer treatment.

## 1. Introduction

Cancer is a global health issue that brings many medical challenges for patients. It was estimated in 2018 that 17 million new cases of cancer had been diagnosed and over half of these patients ended up dying. In addition, the reports show that the new cases would rise up to 27.5 million by 2040 [1]. World Health Organization (WHO) defines cancer as the second leading cause of mortality, leading to one in six deaths worldwide. These estimations rely on the recent advancements in biotechnology and diagnostic methods. More recent progressions in the field of molecular biology and genetics have also allowed for a better understanding of the disease. Therefore, today the word “cancer” could be attributed to many disease types that have similar fundamental characteristics [2]. However, Hanahan and Weinberg defined cancer as a genetic disease caused by DNA mutations leading to uncontrolled cell proliferation [3]. Their description of cancer relies on the somatic mutation theory and its cell-based gene variations. According to their point of view, cancers have some basic characteristics, which have called “hallmarks of cancer”’ [4]. These hallmarks include six biological properties that a tumor may acquire during its developmental process. Indeed, the hallmarks are principles that help to rationalize the intricate nature of neoplastic disease. They include continuous proliferative signals, insensitivity to proliferation inhibitors, escaping apoptosis, uncontrollable replicative capability, increased angiogenesis, and metastatic propagation [3]. It is no clear that the precise explanation of the cancer hallmarks may have translational benefits in the clinical setting, and targeting one or more hallmarks may help to defeat cancer or increase patient survival [5]. As mentioned before, cell death evasion is one of the major hallmarks of cancer. The physiologic activity of apoptosis is to maintain a balance between cell death and cell proliferation.

Apoptosis is a multistep process that involves two major pathways to trigger a cascade of events leading to the fragmentation of chromatin and nuclear membrane. However, when this physiological process tended to be dysregulated, many pathological transformations happen to develop cancer [6]. Thus, seeking a way to induce this process in cancer cells may have beneficial effects in hampering cancer development and growth. Plant extracts and plant-derived natural molecules are such promising candidates for use as anticancer therapeutics. In some parts of the world, plant species have been reported to treat some cancer types, such as prostate, pancreas, stomach, oral, cervix, breast, colon, lung, hepatic, skin, and blood cell malignancies [7,8,9,10]. Further analysis of in vitro studies illustrated that anticancer properties of the secondary metabolites in the plant extracts are exerted through DNA damage and induction of apoptosis in cancer cells [10]. This review article briefly describes various mechanisms of apoptosis and its deregulation during malignant transformations and summarizes the plant materials with the capacity to target both pathways of apoptosis in cancer cells.

## 2. Major Apoptotic Pathways

The term “apoptosis” is derived from a Greek word with the meaning of falling off of the dead leaves from trees in autumn [11]. Kerr et al. first proposed this term to describe morphological alterations of the cells during this process. These morphological changes occur through two main steps. The early apoptosis events include pyknosis (i.e., decreased cell volume), nuclear degradation, and chromatin condensation [12]. At the later step of apoptosis, initial morphological changes in the cells include cell shrinkage, plasma membrane blebbing, cytoplasmic organelles modification, loss of cell membrane integrity, and production of apoptotic bodies [13]. The process of apoptosis was identified very late in the history of cellular biology; because the apoptotic cells are usually engulfed by phagocytes before the formation of apoptotic bodies. The presence of apoptotic bodies was discovered in vitro under particular conditions. Under these conditions, the remnants of apoptotic cells generally undergo degradation, which is known as secondary necrosis [14].

Early in apoptosis, phosphatidylserine (PS) molecules usually are flipped out to the outer layer of the cell membrane. PS exposure to the cell surface displays an engulfment signal, which attracts phagocytic cells to engulf apoptotic cells without the secretion of inflammatory cytokines [15]. It has been clearly described that various endogenous and exogenous agents trigger programmed cell death in a specific cell type. Physical stimulators (such as radiation, trauma, and chemotherapeutics) and infectious pathogens (viruses and bacterial toxins) are exogenous factors that affect most types of cells. Endogenous activators of apoptosis include the absence of growth factors, trophic hormone deficiency, glucocorticoid therapy, and ablation of matrix attachment [16]. Regardless of the type of stimulators of apoptotic cell death, this process is usually triggered through two distinct mechanisms and pathways that are completely explained in the next section. Figure 1 depicts major apoptotic pathways in relevant factors.

### 2.1. Extrinsic Pathway of Apoptosis

The first pathway of apoptosis, named the extrinsic pathway, is triggered via cell-surface proteins that are known as death receptors. Initiation of this pathway occurs when death receptors, such as Fas, the tumor necrosis factor (TNF) receptors TNFR1 and TNFR2, and the TNF- related apoptosis-inducing ligand (TRAIL) receptors DR4 and DR5 are occupied by Fas ligand, TRAIL, and TNF [17,18,19,20]. The intracellular parts of the death receptors have a conserved protein–protein interaction domain known as the death domain that is binding sites for adaptor proteins, such as TNF receptor-associated death domain (TRADD) and Fas-associated death domain (FADD), as well as initiator caspases like caspase 8 and 10 [21,22,23]. In this pathway of apoptosis, the cellular FADD-like IL-1β-converting enzyme (FLICE) inhibitory protein (c-FLIP) acts as an inhibitor by negative regulation of the activation of these procaspase proteins [24]. Activated caspases 8 and 10 subsequently activate effector caspases (3, 6, and 7) and cleave BH3 interacting-domain death agonist (BID) protein, which translocates to the mitochondria and causes the release of cytochrome C into the cytoplasm [25]. Therefore, the extrinsic pathway can trigger apoptosis through direct activation of effector caspases or by mitochondria-dependent pathway.

This interesting activity of BID depends on the type of the cell; in type II cells (which are mainly dependent on the intrinsic pathway), BID helps to fortify the apoptotic signal by involving the mitochondria and causing more effective apoptosis in these cells., but in type I cells (which mainly trigger the extrinsic pathway), BID action is not necessary, and death receptors efficiently trigger apoptosis by activating downstream caspases [26]. In addition to FLIP protein, there are cell-surface proteins called decoy receptors (DcRs) that act as a negative regulator of the extrinsic pathway of apoptosis. These receptors compete with the death receptors to bind death ligands and thereby limit apoptosis signaling through death receptors. DcR1 and DcR2 have been shown to bind TRAIL receptors and DcR3 acts as a decoy receptor for FasL in several cancer cells [27].

### 2.2. Intrinsic Pathway of Apoptosis

As is implied from its name, the intrinsic pathway of apoptosis is triggered by the stimulators from intracellular space. Furthermore, this pathway is known as intrinsic because it is mediated mainly by mitochondria within the cell [28]. Many stimulators, such as oxidative stress, irradiation, cytotoxic drugs, DNA damage, and hypoxia, are involved in the activation of the intrinsic pathway of apoptosis [29,30,31]. Regardless of the cause, the pivotal element for the intrinsic pathway’s initiation is augmented mitochondrial outer membrane permeability and the release of cytochrome C from the mitochondrial intermembrane space into the cytoplasm [32]. The release of cytochrome C from mitochondria occurs via several possible mechanisms, including the induction of mitochondrial permeability transition (MPT), proapoptotic B-cell lymphoma protein 2 (Bcl-2) family proteins (like Bax/Bak), and decreased hypotonicity of cytoplasm due to ionic effluxes [33]. In the cytosol, the released cytochrome C forms a multiprotein complex structure with apoptotic protease activating factor-1 (Apaf-1) and procaspase-9, which is known as the apoptosome.

Apoptosome complex plays an important role in converting procaspase-9 to active caspase-9, which in turn contributes to the activation of effector caspases signaling, leading to the destruction of the cell by the apoptosis process [34]. As mentioned before, the Bcl-2 family members are the major regulators of the intrinsic pathway of apoptotic cell death. This family is comprised of two groups of proteins with opposite activities in the apoptosis process. For example, Bax, Bak, Bad, Bcl-Xs, Bid, Bik, Bim, and Hrk play proapoptotic activity, but Bcl-2, Bcl-xL, Bcl-W, Bfl-1, and Mcl-1 have an antiapoptotic role [35]. Proapoptotic proteins act by forming mitochondrial membrane pores to release cytochrome C, but antiapoptotic ones function oppositely to block their action [36]. Apoptosis-inducing factor (AIF), the second mitochondria-derived activator of caspase (Smac)/direct IAP binding protein with low pI (DIABLO), and Omi/high-temperature requirement protein A (HtrA2) are other apoptotic proteins that are released from the mitochondria during cytochrome C efflux. These three proteins act by binding to the inhibitor of apoptosis proteins (IAPs) and prevent them from interacting with and inhibit caspase-9 and -3 during the apoptosis process [37].

### 2.3. The Mechanisms of Apoptosis Evasion in Cancer

One of the main hallmarks of cancer is cell death evasion [3]. Several stressor stimuli induced by cancerous conditions compel cancer cells to avoid apoptosis, leading to the development of a tumor and resistance to therapy [38]. Cancer cells elaborate several mechanisms to dysregulate intrinsic or extrinsic pathways to evade apoptosis (Figure 2).

The first molecules that are impaired in the extrinsic pathway are death receptors. For instance, downregulated CD95, as a death receptor, has been attributed to the resistance of leukemia and neuroblastoma cell lines to treatment approaches [39]. Furthermore, defective molecules that act downstream of the death receptors may contribute to the development of apoptosis-resistance in tumors.

Tourneur et al. found that downregulation or loss of expression of the FADD protein in leukemic cells was a predictor of resistance to chemotherapy and a poor prognostic factor [40]. Treatment modalities targeting TRAIL receptors revealed that the main cause of apoptosis resistance in colon cancer cell lines is the inappropriate transport of DR-4 and DR-5 receptors from intracellular sources to the cell surface [41]. Further, epigenetic mechanisms like CpG-island hypermethylation can hamper the expression of death receptors and lead to the decreased levels of the receptors in the cell membrane [42]. The DcRs are another way for cancers to evade apoptosis. For example, DcR3, which binds Fas ligand by a competitive mechanism and blocks Fas ligand-mediated apoptosis, was identified to be upregulated in glioblastoma and carcinomas of lung and colon [43]. In the meantime, upregulation of c-FLIP, an antiapoptotic factor, has been shown in numerous cancers [44,45,46]. caspase-8 is another target molecule modified in a variety of cancers. Neuroblastoma and carcinomas of colorectal and head and neck tissues harbor inactivating genetic mutations of caspase-8 [47,48,49].

Epigenetic mechanisms are also implicated in the inactivation of caspase-8 during tumor development and progression [50,51]. A number of potential mechanisms have also been proposed for cancer cells to evade apoptosis through targeting the intrinsic pathway. Overexpression of the antiapoptotic BCL-2 family proteins and underexpression of proapoptotic factors, such as Bax, has been numerously reported in various cancers [52,53]. This is the predominant mechanism for the escape of cancer cells from apoptosis. Limiting the release of cytochrome C is another method to circumvent programmed cell death during cancer development. Neuroglobin (NGB) is an oxygen-binding globin protein that forms a complex with cytochrome C, preventing its release into the cytosol and activation of the caspase 9 [54]. NGB has been reported to be overexpressed in cancer cells rendering them chemo- and radiotherapy resistance [55]. Apaf-1 is another target with altered expression or activity in some types of cancer cells. These alterations have been described to block apoptosis cascade and give the ability to a cancer cell to survive and resist treatment [56,57]. Aberrant expression or function of IAP proteins is also a way used by cancer cells to avoid programmed cell death. For example, the X-linked inhibitor of apoptosis protein (XIAP) is frequently upregulated in many cancer tissues and cells and has been recognized to be responsible for the cancer cell’s resistance to various apoptotic stimuli [58].

### 2.4. Plant Materials That Simultaneously Target Both Intrinsic and Extrinsic Pathways

#### 2.4.1. Plant Extracts

According to their ancestral use in traditional medicine, various plant extracts have been shown to activate apoptosis in human cancer cells (see Table 1).

The ethanolic extract from *Azadirachta indica* (known as neem) leaf has been discovered to suppress the growth of squamous cell carcinoma in a hamster model of the disease. This extract acted by the induction of apoptosis through elevating the expression of proapoptotic factor Bim and caspase-8 and -3 as well as the downregulation of Bcl-2 expression, suggesting the modulation of both intrinsic and extrinsic pathways due to the exposure of the tumor to the extract [59]. Antiproliferative effects of *Camellia sinensis* (known as tea plant) on the colorectal cancer cell line, HT-29, was affirmed to be attributed to its apoptosis-triggering capability, which is evidenced by activating caspase-3, -8, and -9 in these cancer cells. This indicates that white tea leaves extract may act by orchestrating both death receptor- and mitochondrial-mediated apoptosis in colon cancer [60]. Using *n*-hexane fractionation prepared from ethanolic extract of the roots of *Inula racemosa* (known as pushkarmool), Pal and colleagues showed that treating a human leukemia cell line, HL-60, by this extract induced both pathways of apoptosis. This product enhanced the activity of caspase-9, -3, and -8 but remarkably decreased mitochondrial membrane potential (MMP), cytochrome C release, and Bax translocation to the mitochondrial membrane [61].

Evaluation of the effect of ethyl acetate extract of *Uncaria tomentosa* (known as Cat’s Claw plant) on HL-60 cells demonstrated that this plant extract act not only by triggering intrinsic apoptotic pathway through collapsing MMP, decreasing Bcl-XL, and increasing Bax, cytochrome C efflux, and caspase-9 activation, but also by elevating the membrane-bound Fas in addition to the activation of caspase-8 and cleavage of Bid [62]. Treating human hepatoma cell lines (HepG2 and HA22T/VGH cells) with ethanolic extract of *Cinnamomum kanehirai* (known as small-flowered camphor tree, or stout camphor tree) leaves unraveled its apoptosis-inducing activity, which exerted by targeting the cleavage of caspase-3 and enhancement of caspase-8 and caspase-9 activity. Additionally, this treatment significantly elevated the ratio of Bax to Bcl-2 [63]. Human cervical carcinoma HeLa cell death has been reported to be induced by exposing them to ethanolic extract prepared from mango peels (*Mangifera indica*). This extract inhibited the expression of Bcl-2 and thereby caused the activation of caspase-3, 7, 8, and 9 in these cancer cells [64]. Chloroform extract prepared from the whole plant of *Solanum lyratum* (known as nightshades) has been shown to stimulate both extrinsic and intrinsic pathways of apoptosis in three human oral cancer cell lines (HSC-3, SAS, and CAL-27 cells). The extract acted by downregulating Bcl-2 and Bcl-xl but upregulating Bax and Bad in these cell lines [65]. It also induced ROS formation, decreased MMP, and activated caspase-8, -9, and -3 [65].

Dichloromethane extract of the roots of *Toddalia asiatica* (known as an orange climber) has shown the strongest antiproliferative activity compared to other fractions. Treating HT-29 human colon cancer cell line with this plant product promoted both intrinsic and extrinsic pathways of apoptosis by enhancing the activity of caspases -8, -9, and -3. Moreover, ROS generation appears to be the main reason for the cell cycle arrest in these cells following extract exposure [66]. Ethanolic extract of the rhizome of *Cyperus rotundus* (known as coco-grass, Java grass, nutgrass, purple nutsedge or purple nutsedge, red nut sedge, Khmer kravanh chruk) suppresses the proliferation of triple-negative human breast cancer cell line, MDA-MB-231. This plant material mainly acts by simultaneous activation of both pathways of apoptotic cell death in these cells. On one hand, the plant extract upregulates DR5 and proapoptotic Bax but downregulates antiapoptotic Bcl-2 and survivin. On the other hand, it acts by decreasing Bid expression and activating caspase-8 and -9, -3, but increasing mitochondrial membrane depolarization [67]. MCF-7 breast cancer cell line has been revealed to die due to the exposure to the methanolic extract of *Euphorbia hirta* (known as asthma-plant) whole plant. The extract affected various components of intrinsic and extrinsic pathways of apoptosis through reduction elevation of intracellular ROS, fragmentation of DNA, and activation of caspase-2, -6, -8, -9, and -3 [68]. Exposing the multidrug-resistant human gastric cancer SGC7901/ADR cell line to *n*-hexane extract of the aerial parts of *Euphorbia lunulata* (known as crescent-shaped Euphorbia, leafy spurge or cateye spurge) inhibited the proliferation and invasiveness of the cells. The effect of this extract on the inhibition of proliferative capacity of SGC7901/ADR cells was accompanied by the induction of apoptosis through increased activities of caspase-3, -8, and -9 as well as the overexpression of Bax, underexpression of Bcl-2, and release of cytochrome C into the cytoplasm of the cells [69].

The leaf extract of *Hibiscus sabdariffa* (known as Roselle) induced intrinsic and extrinsic apoptosis in androgen-dependent prostate cancer cell lines (LNCaP cells). This medicinal herb extract declined the expression of Bcl-2 and collapsed MMP but increased cytoplasmic cytochrome C, FasL, Bax, *t*-Bid levels and activities of caspase-3, -9, -8 [70]. In vivo data from this study indicated that treating nude mice xenograft model of this cancer with Roselle leaf extract (50 or 100 µg/mL) not only decreased tumor burden but also increased the expressions of FasL, Bax, and cleaved caspase-3 in tumor tissues [70]. The chloroform fraction of the extracts of the stems and leaves of *Narcissus tazetta* var. *chinensis* (known as Chinese Sacred Lily) has been identified to trigger both intrinsic and extrinsic apoptosis processes in the HL-60 cell line. This herbal product influenced apoptosis pathways via upregulation of Bax, downregulation of Bcl-2, the subsequent release of cytochrome C into the cytosol, and the activation of caspase-8, -9, and -3 enzymes [71]. Sitarek et al. used methanolic root extract of *Leonurus sibiricus* (known as honeyweed or Siberian motherwort) to evaluate its anticancer effects on human glioma primary cells. They experimentally illustrated the apoptosis-inducing activity of the extract, which was confirmed by the production of ROS, loss of MMP, and cell cycle arrest in these cells. Then, they found that this treatment caused the elevation of the mRNA and protein levels of Bax, p53, caspase-3, -8, and -9 while declining the expression of Bcl-2 [72]. Qingjie Fuzheng granule (QFG) is comprised of four Chinese medicinal plants, including *Hedyotis diffusa* Willd, *Malt*, *Astragalus*, and *Scutellaria barbata* D. Don. Zhong et al. conducted an investigation to assess QFG effects on the growth of hepatocellular carcinoma (HCC) cells both in vivo and in vitro. The results showed that QFG hampered cell proliferation by upregulating Fas, FasL, and Bax and downregulating Bcl-2 protein. It also induced the activation of caspase-8, -9, and -3. In vivo study of this poly-herbal formulation on HCC xenograft mice showed an inhibitory effect of QFG on HCC tumor growth without any toxicity [73].

Studies of Yong et al. on QFG anticancer activities against colorectal cancer cell lines HCT-116 and HCT-8 also led to the same data [74]. The exposure of MCF-7 cells to methanol and butanolic extracts of *Oldenlandia diffusa* (known as *Oldenlandia diffusa* (Willd) Roxb) has been discovered to activate caspase-8, and caspase-7, while induced Bax expression and reduced Bcl-2 level [75]. Hwang-Heuk-San (HHS) is a mixture of some Korean medicinal herbs with anticancer activity against HCT116 human colorectal cancer [76]. This herbal formula influenced the intrinsic pathway of apoptosis by inducing ROS production, leading to the decreased MMP and increased cytochrome C levels in the cytosol. The mixture also amplified Bax, reduced Bcl-2 levels, and activated caspase -9. It also acted on the extrinsic pathway through the activation of Bid and caspase-8 together with overexpression of FasL, DR4, and DR5. Finally, the effector caspase (caspase-3) was stimulated to cleave and activate other components of the cascade [76]. So-Cheong-Ryong-Tang is another polyherbal mixture whose fermented product induced apoptosis in AGS gastric adenocarcinoma cell line by activating the caspase-3, -8, and -9. Testing this polyplant mixture in murine models of gastric adenocarcinoma led to significant inhibition of tumor weight 48.6% compared to control animals [77]. The positive staining for anti-Bcl-2, -Bax, -PARP, -caspase-8, and -caspase-3 antibodies in SNU-16 gastric carcinoma cells following their exposure to chloroform extract of guava leaves (*Psidium cattleianum*) uncovered the involvement of intrinsic and extrinsic pathways in apoptosis-inducing effects of this natural product [78].

The chloroform extract of *Cucurbita ficifolia* (known as fig-leaf gourd, Malabar gourd, black seed squash and cidra) fruit acted on MCF-7 cells through targeting apoptotic pathways. It amplified the expressions of FADD, BAK, BAX, and caspase-8, -9, -3 [79]. *U*-2OS human osteosarcoma cells underwent apoptotic cell death due to the exposure to crude extract of *Corni fructus* (known as Shan Zhu Yu in China). This plant extract elevated both activity and protein level of caspase-8, -9, and -3. The formation of ROS, overproduction of intracellular Ca^2+^, reduction of MMP, and also increased levels of Bax, cytochrome C, AIF, Fas, and TRAIL in the apoptotic cascade were observed following this treatment, indicating the concurrent induction of both receptor and mitochondrial-mediated apoptosis in this cell line [80]. Ethanolic extract of the fruits of *Brucea javanica* (known as Macassar kernels) has been observed to have apoptogenic effects on HT29 colon cells. Its mechanism of action involves both receptor- and mitochondria-mediated pathways. The extract amplified some key factors in the extrinsic pathway, including Fas, TNFR1, TNF2, DR6, CD40, Bid, caspase-8, and TRAIL-4. It also acted on the intrinsic pathway by upregulating Bax, Bad, cytochrome-c, and downregulating Bcl-2 along with the activation of caspase-9 [81]. Overall, the literature contains a large number of studies about plant extracts able to induce apoptosis in human cancer cells through activation of molecules of the intrinsic and/or extrinsic apoptosis pathways. Importantly, some of these crude plant extracts were further found active in vivo, causing cancer cell apoptosis and reducing cancer progression. Although these observations conducted with crude plant extracts in vitro/in vivo support the ancestral use of these plants in traditional medicine to fight cancer, occidental medicine and science require confirmation of such effect with pure molecules isolated from plants.

#### 2.4.2. Isolated Phytoconstituents

Numerous plant-derived molecules have been shown to activate apoptosis (Table 2 and Figure 3).

Three cancer cell lines (DU-145, PC-3, A-549 cells) were exposed to various concentrations of nimbolide, which is a triterpene found in neem tree. The data showed that nimbolide significantly activates caspase-3, -8, and -9 compared to untreated control cells. This indicated that nimbolide potently triggers both extrinsic and intrinsic apoptosis [82]. Anthocyanins are classic flavonoid compounds with various medicinal effects. Anthocyanin constituents isolated from *Vitis coignetiae* (known as crimson glory vine) kill human leukemia U937 cells by stimulating the apoptosis process in these cells. This effect is exerted through activating Bid and caspase-3, -9, and -8 along with the elevation of Bax and diminution of MMP, Bcl-2, XIAP, cIAP-1, cIAP-2. However, U937 cells with higher levels of antiapoptotic Bcl-2 protein do not respond to the apoptosis-inducing effects of anthocyanins [83]. Phytosphingosine, a phospholipid found in various organisms, such as plants, fungi, and animals, is a key factor involved in many cellular processes. Park et al. showed that phytosphingosine promoted apoptotic cell death in human T-cell lymphoma, Jurkat, and human non-small cell lung cancer cells, NCI-H460. This phospholipid activated caspase-8 via an unknown mechanism without modulating death receptors. Additionally, phytosphingosine induced translocation of Bax into mitochondrial membrane and loss of MMP, which led to cytochrome C release and subsequent activation of caspase-9 and -3 without the involvement of caspase-8 in this process [84].

A derivative of phytosphingosine, named N, N-dimethyl phytosphingosine (DMPS), also affects both intrinsic and extrinsic pathways in human leukemia cells (HL-60 cells). However, activation of caspase-8 is the pivotal step in the induction of the intrinsic pathway, which is accompanied by impairment of the MMP, release of cytochrome C, activation of caspase-9 and caspase-3, and suppression of the antiapoptotic members of the Bcl-2 family [85]. Tubeimoside-1, a triterpenoid saponin, stimulates both pathways of apoptosis in the human hepatoma cell line (HepG2 cells). Tubeimoside-1significantly elevated the levels of cleaved caspases-3, -8, and -9 and the expression of Fas, FasL, and Bak, but downregulated Bcl-2 protein levels in HepG2 cells. Interestingly, tubeimoside-1 failed to alter the expression level of Bax protein [86]. Tetrandrine (TET) and cepharanthine (CEP), as two alkaloid compounds found in medicinal plants, are reported to function as apoptosis-inducing agents. Xu et al., by treating human leukemia Jurkat T cells with the combination of these alkaloids, have declared that they significantly upregulated both initiator caspases (caspase-8 and 9) and effector caspases (caspase-3 and 6). TET and CEP also enhanced the expression of Bax and p53 but negatively affected the expression of Bcl-2 and Mcl-1 [87].

Berberine, as an isoquinoline alkaloid of the protoberberine type, induced apoptosis of SCC-4 human tongue cancer cells by activating expression of caspase-8, -9 and -3, AIF, and endoG. It also increased the ratio of Bax/Bcl-2 and altered MMP in these cancer cells [88]. Wogonin, a flavonoid-like compound found in *Scutellaria baicalensis* (known as Baikal skullcap or Chinese skullcap), has been discovered that triggered apoptosis of U-2OS human osteosarcoma cells by augmenting the expression of both intrinsic and extrinsic apoptosis components, including Bax, Bad, cytochrome C, cleaved caspase-9, cleaved caspase-3, AIF, Endo G, Fas, caspase-8, and caspase-4 [89]. Anticancer effect of wogonin in vivo was shown in athymic nude mice inoculated with two breast cancer cells (T47D and MDA-MB-231 cells). The results revealed a decreased xenografts burden by up to 88% without any toxicity after 4 weeks of treatment [90].

Treating *U*-2OS cells with ouabain (a steroid or cardiac glycoside) resulted in the activation of caspase-3, -8, and -9, the elevation of Bax, AIF, Endo, cytochrome C release, but downregulation of Bcl-2. Ouabain also decreased the levels of ROS and MMP while increased Ca^2+^ levels in *U*-2OS cells [91]. Xenografting neuroblastoma SH-SY5Y cells into immune-deficient mice treated with 2 mg/kg/day ouabain significantly decrease tumor volume and activated caspases-3 enzyme in these tumor tissues [92]. Kim et al. conducted an investigation to study the potential of saikosaponin A (a triterpenoid saponin) in inducing death of human colon carcinoma cell lines. Their data unraveled that saikosaponin A promoted apoptosis by affecting death receptor and mitochondrial-dependent processes, which converged on the elevated activity of caspase-2, -8, and -9 as well as the activation of Bax and Bid proteins [93]. Yan et al. chemically modified the structure of emodin to an emodin azide methyl anthraquinone derivative (AMAD), which is isolated from the rhizome of *Polygonum sachalinense* (known as giant knotweed. Then, they experimentally demonstrated AMAD-induced apoptosis of human breast cancer cell cells (MDA-MB-453 cells) and human lung adenocarcinoma cells (Calu-3 cells). They further discovered that AMAD dropped MMP, caused the efflux of cytochrome C, activated caspase-8, -9, and -3. They also confirmed that the mitochondrial-mediated promotion of apoptosis was probably dependent on caspase-8 activation and subsequent cleavage of Bid protein [94]. In vivo data on the anticancer effects of emodin (40 mg/kg/day) revealed its suppressing capacity against tumor weight of nude mice xenografts bearing LS1034 colon cancer cells [95].

A triterpenoid compound isolated from *Anemone raddeana* (known as Toujian Liang in China), raddeanin A, has been suggested to trigger the apoptosis process in three different gastric cancer cells (BGC-823, SGC-7901, and MKN-28 cells). The molecular biology techniques clarified that raddeanin A amplified Bax expression while diminished the expressions of Bcl-2, Bcl-xL, and survivin. This terpenoid also augmented the activities of caspase-3, -8, -9, and poly-ADP ribose polymerase (PARP) enzymes in all three cell lines [96]. Raddeanin A shows strong antitumor activities in syngeneic models of granuloma cell line S180, hepatic carcinoma cell line H22, and cervical cancer cell line U14. Although different concentrations of raddeanin A (0.5, 1.5, and 4.5 mg/kg) were applied for the treatment of these mice models, 4.5 mg/kg dose of raddeanin A more significantly inhibited the volume of these tumors [97].

Celastrol, as a pentacyclic triterpenoid isolated from *Tripterygium wilfordii* (known as thunder god vine), acted on the death receptor and mitochondrial-dependent pathways to initiate apoptosis in human non-small-cell lung cancer cell line, A549. In the intrinsic pathway, celastrol promoted the release of cytochrome C from mitochondria and induced the upregulation of Bax and downregulation of Bcl-2 proteins. In the extrinsic pathway, it stimulated the expression levels of Fas and FasL. Consequently, celastrol led to the proteolytic activation of caspase-9, -8, -3, and PARP protein [98]. Celastrol also affected osteosarcoma cells (HOS and MG-63 cells) apoptosis process through elevating the activity of caspase-3, -8, and -9 along with the upregulation of DR5 and proteolytic activation of Bid [99]. This terpenoid has the potential to be used as an anticancer agent in vivo. Celastrol has been described to inhibit the growth of human glioma xenografts in a murine model of this cancer through suppressing angiogenesis [100]. Arctigenin, bioactive lignin isolated from some plants, triggers apoptosis in HepG2 and SMMC7721 hepatocarcinoma cells. This compound affects the mitochondrial pathway by inducing loss of MMP, the elevation of Bax, diminution of Bcl-2, an increase of cytochrome C in the cytosol, and activation of caspase-9 and -3. Furthermore, it influences the extrinsic pathway by upregulating the levels of Fas/FasL and activating caspase-8 [101].

Oleandrin, a toxic glycoside, induces death in two osteosarcoma cell lines (*U*-2OS and SaOS-2 cells) by acting on apoptotic cell death via the generation of intracellular ROS and loss of MMP which leads to the release of cytochrome C into the cytoplasm. Oleandrin also reduced the Bcl-2 level. However, it induced the expressions of Bax, Fas, FasL, caspase-9, -8, -3, and activity of caspase-3 in these cancer cells [102]. Casticin is a flavonoid compound that can induce apoptosis in colon cancer cell lines (HT-29, HCT-116, and SW480 cells). Tang et al. uncovered that treating these three cells with casticin lowered the expression of antiapoptotic Bcl-2 and Bcl-xL, inhibitors of apoptosis-like Bax, XIAP, and cFLIP. This flavonoid also incremented the expression of DR5 while had no effect on other cell surface receptors, such as DR4 and DcRs [103]. Lambertianic acid (LA) is a phytoconstituent isolated from *Pinus koraiensis* (known as Korean pine) leaves with the capacity to initiate apoptosis in non-small cell lung cancer cells (A549 and H1299 cells). A study by Ahn and colleagues showed that this active compound in combination with TRAIL orchestrated the induction of both pathways of apoptosis through, which Bcl-2, FLIP, and XIAP were decreased, but Bid and caspase-3, -,8, -9 were activated and DR4 was upregulated in both cell lines [104]. Experimental evidence obtained from the treatment of H460/R cells (resistant non-small cell lung cancer cell line) with the combination of galbanic acid, a sesquiterpene coumarin, with TRAIL established apoptosis-promoting activity for this combination therapy. This activity was observed as the activation of caspase-9, caspase-8, and PARP along with overexpression of DR5 and underexpression of Bcl-2, Bcl-xL, and XIAP in H460/R cells [105].

The combination of apigenin, a flavonoid, with TRAIL also has antiproliferative and apoptogenic impacts on A549 and H1299 cells. Chen and colleagues provided evidence to prove that this combination therapeutic strategy showed a synergistic effect on the upregulation of death receptors (DR4 and DR5) in the extrinsic pathway and on the amplification of Bad and Bax as well as the inhibition of Bcl-xL and Bcl-2 in the intrinsic apoptosis pathway [106]. Their in vivo study conducted to evaluate the effects of apigenin on the growth of A549 xenografts in mice models of the disease indicated that apigenin not only inhibits tumor growth but also increases the expressions of DR4 and DR5 receptors [106]. Exposing human ovarian cancer cells OVCAR-3 and SKOV-3 to kaempferol, a flavonoid, caused a significant amplification in the expression levels of proapoptotic factors, such as Bax, caspase-3, -8, and -9, while decreased antiapoptotic proteins, including Bcl-2, Bcl-xL, survivin, XIAP, c-FLIP in these cells. Kaempferol, in combination with TRAIL, also showed the same results as kaempferol alone [107]. Rosamultic acid, a triterpenoid found in *Rosa multiflora* (known as rambler rose and baby rose) roots, has been revealed as a natural anticancer product against the human gastric cancer cell line (SGC-7901 cells). Rosamultic acid-induced cell cycle arrest and apoptosis in SGC-7901 cells through cleaving PARP and activation of caspase-3, -8, and -9, indicating the promotion of receptor and mitochondrial-dependent apoptosis due to rosamultic acid treatment [108].

As a plant-derived flavonoid, acacetin kills gastric cancer cells (AGS cells) by initiating an apoptosis cascade in them. The molecular mechanism of this effect involves both intrinsic and extrinsic pathways. Acacetin affects the intrinsic pathway by producing ROS, collapsing MMP, augmenting Bax and p53, declining Bcl-2, activating Bad, inducing the efflux of cytochrome C from mitochondria, and subsequent activation of caspase-9. However, it promotes the extrinsic pathway via increasing the expressions of Fas and FasL and activation of caspase-8 and Bid proteins [109]. Lycorine is an alkaloid compound isolated from the Amaryllidaceae family with the capacity to suppress the growth of HL-60 leukemia cells and cause its apoptosis. Lycorine-treated cells show an increased ratio of Bax/Bcl-2 proteins along with the higher activities of caspase-8, -9, -3 enzymes compared to untreated cells, which explains the implication of intrinsic and extrinsic apoptosis pathway in this effect of lycorine [110]. Lycorine derives a similar effect when is exposed to the KM3 human multiple myeloma cell line [111]. HL-60 cells underwent apoptotic cell process when they were treated with various concentrations of meisoindigo. Meisoindigo, as an active constituent of *Indigo naturalis* (known as Qing dai in China), elevates the activities of caspase-3, -8, -9, and PARP, while reducing the levels of Bcl-2 and inducing the expression of Bax, Fas receptor and the release of cytochrome C into the cytosol of HL-60 cells. This implies that both cell surface receptors and mitochondria are implicated in the apoptosis-inducing capacity of meisoindigo [112]. Methyl ferulate is a derivative of the *Tamarix aucheriana* plant (known as salt cedar) that has been shown to trigger apoptosis in two colorectal cancer cell lines (SW1116 and SW837 cells). To initiate apoptotic cell death, methyl ferulate increased the expression of Bax, Bad, Apaf1, Bid, Bim, Smac, and various initiator and effector of caspases, including caspase-2, -3, -6, -7, -8, and -9. It also downregulated antiapoptotic proteins involved in the apoptosis, such as c-IAP-1, c-IAP-2, Bcl2, and FLIP [113].

Thiosulfinates are esters isolated from *Allium tuberosum* (known as garlic chives) that previously have been discovered to have anti-growth effects on PC-3 human prostate cancer cells. Kim et al. identified their apoptogenic activity, which was observed as the augmented activities of caspase-8, -9, and -3, as well as the cleavage of Bid, downregulation of Bcl-2, and upregulation of Bax and AIF [114]. A study by Lee et al. conducted to evaluate the proapoptotic effects of thiosulfinates in human colon carcinoma cell lines (HT-29 cells) also elicited similar results [115]. RC-58 T/h/SA#4 prostate cancer cells treated with *Corni fructus* (known as Shan Zhu Yu in China) -derived triterpene, ursolic acid, showed apoptotic features, including nuclear condensation, apoptotic body formation, and DNA fragmentation. Ursolic acid also altered various apoptosis pathway components, such as the activation of caspase-3, -8, -9, and Bid along with the overexpression of Bax and reduction of Bcl-2 proteins, suggesting the promotion of intrinsic and extrinsic apoptosis in these cancer cells [116]. A study by Kim et al. provided evidence to prove the anticancer impact of crude saponins extracted from *Platycodon grandiflorum* (known as Kilkyong in Korea, Jiegeng in China, and Kikyo in Japan) roots against HT-29 cells. Their further tests demonstrated that these phytochemicals induced apoptosis through activation of PARP, Bid, and caspase-8-, -9, and -3 along with the fragmentation of DNA, augmentation of Bax level, and reduction of Bcl-2 protein [117].

Fisetin is a dietary flavonoid, which uses an unusual mechanism to induce apoptosis in caspase-3-deficient MCF-7 cells. It decreases MMP along with activation of caspase-7, -8 and -9, and PARP cleavage but has no obvious effects on DNA fragmentation and phosphatidylserine cell surface exposure [118]. Fisetin can activate both caspase-8 and caspase-9 in triple-negative breast cancer cell lines, including MDA-MB-468 and MDA-MB-231 cells [119]. This flavonoid affected apoptosis pathway components in prostate cancer LNCaP cells by increasing cytochrome C release, downregulating Bcl-2 and XIAP, and activating caspases-3, -8, and -9 [120]. Treatment of Lewis lung carcinoma cells (LLC)-bearing mice with fisetin inhibited the tumor growth by 67% compared to the 66% inhibition rate produced by low-dose cyclophosphamide. Fisetin seemed to exert this effect by hampering angiogenesis in tumor tissues [121]. (-)-Anonaine is a bioactive alkaloid that has been proven to cease the proliferation of human cervical cancer (HeLa) cells by inducing apoptotic pathways in them. This alkaloid acted on the apoptotic process by increasing intracellular ROS, disrupting MMP, upregulating Bax, and activating caspase-3, -7, -8, and -9 in these cancer cell lines [122]. A flavonoid compound, ampelopsin, has been experimentally affirmed that could activate both pathways of apoptotic cell death via raising the activities of initiator caspases-8 and -9 and the effector caspase-3 in two human glioma cell lines (U251 and A172 cells) [123]. Ampelopsin prevented tumor growth of human glioma xenograft in vivo. It affected apoptosis of these tumor cells by activating caspase-3, -8, and -9, as well as the enhancement of PARP expression [123]. Britannin, a sesquiterpene lactone, triggers apoptosis of SMMC-7721 and HepG2 liver cancer cell lines through modulating both the extrinsic and intrinsic pathways by the activation of caspase-8, -9, and -3 [124]. Treating HepG2 bearing male BALB/c nu/nu nude mice with britannin significantly hampered the growth of these tumors in vivo. Further experiments showed that britannin downregulated both ki-67 and phosphorylated-mammalian targets of rapamycin (mTOR), whereas upregulated phosphorylated-AMP-activated protein kinase (AMPK), cleaved caspase-3, and light chain 3 (LC3) on these tumors [124].

Human cervix adenocarcinoma cells, HeLa, treated with corosolic acid, a pentacyclic triterpene acid isolated from various plants, show an increased ratio of Bax/Bcl-2, reduces MMP, leading to the efflux of cytochrome C from mitochondria and activation of caspase-9 and -3. It also promotes the activation of the extrinsic pathway by augmenting the activity of caspase-8 [125]. *Artemisia princeps Pampanini* (known as Korean wormwood, Korean mugwort, and Japanese mugwort) possesses a natural flavonoid, named eupafolin, with significant anticancer properties. Eupafolin anticancer effects on HeLa cells are observed as Bcl-2 gene alteration, mitochondrial-dependent events, and the activation of caspase-3, -6, -7, -8, and -9 [126]. Dehydrocostus lactone isolated from *Saussurea lappa* root (known as costus) alters various components of the intrinsic and extrinsic pathways to initiate apoptosis in DU145 human prostate cancer cells. These alterations include increased activities of caspases-8, -9, -7, and -3, activation of PARP, downregulation of Bcl-xL, and upregulation of Bax, Bak, Bok, Bik, Bmf, and *t*-Bid [127]. Administration of isoangustone A, as a flavonoid isolated from *Glycyrrhiza uralensis* (known as Chinese licorice) to DU145 cells, promotes extrinsic cascade by amplifying Fas and DR4 as well as the activation of caspase-8 and Bid, while stimulates intrinsic pathway through decreasing MMP and releasing cytochrome C into the cytosol, causing caspase-9 activation. It also activates effector caspases -3 and -7 [128]. Hemanthamine and hemanthidine, alkaloid compounds from Amaryllidaceae family, treatment of Jurkat cells lead to the decreased MMP, strong activation of caspase-9 and caspase-3/7, and weaker activation of caspase-8 [129]. Another alkaloid compound, 6-methoxydihydrosanguinarine, induces apoptosis in HepG2 cells by stimulating the activation of caspase-8, -9, and -3. This alkaloid also increased the ratio of Bax to Bcl-2, leading to the release of mitochondrial cytochrome C into the cytoplasm of HepG2 cells [130].

Sanguinarine is an alkaloid with structural similarities with 6-methoxydihydrosanguinarine. Sanguinarine can trigger apoptosis in primary effusion lymphoma (PEL) cell lines, including BC1, BC3, BCBL1, and HBL6, through orchestrating two apoptosis cascades. It acts by elevating DR5 expression through ROS and promotes the activation of caspase-8 and Bid, which stimulates Bax. This can impair MMP, the release of cytochrome C, and activation of caspase-9 and caspase-3 [131]. A biologically active flavonoid present in the root bark of *Morus australis* known as morusin exerts apoptogenic effects on colorectal cancer HT-29 cells. Morusin targets various factors of the extrinsic and intrinsic pathways, such as the disruption of MMP, inhibition of XIAP, and release of cytochrome C and Smac/DIABLO. In addition, this flavonoid stimulates the activation of caspase-8, -9, and -3 to effectively induce apoptosis in HT-29 cells [132]. Souza-Fagundes and coworkers purified a naturally occurring labdane diterpene, known as myriadenolide, from the leaves of *Alomia myriadenia* and evaluated its apoptogenic activity against two different human leukemia cells (Jurkat and THP-1 cells). Then, they reported that myriadenolide targets both apoptotic pathways via depolarization of mitochondria and activation of Bid, caspase-8, -9 and, -3. However, their further analyses showed that the activation of caspase-8 and Bid was not mediated by death receptors activation, indicating the role of another mechanism in the stimulation of extrinsic pathway components [133]. Plumbagin is a natural naphthoquinone isolated from *Plumbago zeylanica* (known as Chitrak) that effectively targets apoptotic pathways in human promyelocytic leukemia cells (NB4 cells). Plumbagin exerts its effects by increasing Bax and Bak protein levels and decreasing Bcl-xL protein while showed no remarkable effect on Bcl-2 protein. Moreover, plumbagin depolarized mitochondrial membrane and activated caspase-8, -9, and -3 in NB4 cells [134]. Assessment of plumbagin anticancer effects on the growth of xenografts of NB4 cells into NOD/SCID mice showed its suppressing activity against this type of tumor in vivo. Plumbagin reduced the volume of the tumor by 64.49% compared to control animals [134].

*Artocarpus obtusus* has been found to have a xanthone compound called pyranocycloartobiloxanthone A (PA), which shows proapoptotic activity in MCF7 cells. The apoptosis-inducing effect of PA on these cancer cells is mainly observed as the elevation of Bax/Bcl-2 ratio, leading to the collapse of MMP, cytochrome C release, and activation of caspases-9, which in turn induces the activation of effector caspase-3 and -7. PA also exerts these effects through the implication of the extrinsic pathway caspase (caspase-8) [135]. Another xanthone compound, known as α-mangostin, isolated from *Cratoxylum arborescens,* exhibits apoptogenic activity against MCF7 cells. This activity is shown to be exerted through a mechanism that is similar to PA-induced apoptosis [136]. Treating rat models of the xenografted rat LA7 mammary adenocarcinoma cells with 30 and 60 mg/kg α-mangostin for 28 days demonstrated its antitumor activity by reducing tumor volumes by 79.2% for high dose and 74.1% for low dose treatments [136]. Biseugenol B is a natural anticancer compound isolated from *Litsea costalis* that has been evidenced to kill prostatic cancer PC3 cells via apoptosis. Biseugenol B affected the intrinsic pathway by enhancing the ratio of Bax/Bcl-2, causing MMP collapse, the release of cytochrome C, and activation of caspase-9. It also acted on the extrinsic pathway via increasing the level of caspase-8. Biseugenol B also induced the proteolytic activation of executioner caspases-3 and -7 [137]. Shikonin is a bioactive naphthoquinone present in some Chinese medicinal herbs with apoptogenic properties against EL7402 and Huh7 hepatocellular carcinoma cells. Its effects on apoptosis have been illustrated as the proteolytic cleavage of caspase-8, caspase-9, and Bid protein along with diminished protein levels of antiapoptotic factors, such as c-FLIP and Bcl-2. However, it had no obvious effect on the protein level of Bcl-xL [138]. Treatment of Huh7 xenografts in nude mice with shikonin also showed a significantly increased activation of caspase-8, caspase-9, and PARP enzymes, but a significant decrease in tumor volume compared to control animals [138]. Apoptogenic effects of shikonin (Naphthoquinone) in Tca-8113 oral squamous cancer cells involve the downregulation of Bcl-2 protein and proteolytic activation of caspase-8, -9, -3, indicating the promotion of both intrinsic and extrinsic cascades due to shikonin treatment in these cells [139].

### 2.5. Use of Plant Extracts and Plant Molecules in Human Clinical Trials

According to epidemiologic data obtained from a study on the Japanese population, green tea consumption of approximately 10 cups/day significantly decreased the risk of colorectal cancer [140]. Accordingly, Shimizu et al. conducted a clinical trial to evaluate the effects of green tea on 71 patients with previously treated colorectal adenomas. Interestingly, the results showed that the green tea extract drinking (1.5 g per day for 12 months) reduced the incidence of metachronous adenomas and the size of relapsed adenomas in these patients in comparison to the control group [141]. They also reported that no serious adverse side effects were observed in the treatment group than control patients. Another randomized clinical trial was conducted to assess the effect of green tea extract on the prevention of metachronous colorectal adenoma and cancer in the Korean population. The results indicated that consuming the group (0.9 g green tea extract supplementation per day for 12 months significantly reduced the incidences of metachronous adenomas up to 23.6% compared to the control group (42.3%) [142].

A randomized, double-blind, placebo-controlled clinical trial investigated the effects of *Cucurbita ficifolia* (pumpkin) seed oil alone or in combination with saw palmetto oil on 47 patients with symptomatic benign prostatic hyperplasia for 12 months. Pumpkin seed oil (320 mg/day) non-significantly reduced international prostate symptom score and increased quality of life after 3 months of treatment but had no considerable effect on serum PSA compared to control subjects. This oil also gradually improved maximal urinary flow rate after 6 months [143].

A randomized clinical study investigated the effects of *Uncaria tomentosa* extract on the alleviation of the side effects of chemotherapy in patients with breast cancer who were treated with the combination of fluorouracil, doxorubicin, and cyclophosphamide. The data revealed that a combination of chemotherapy and *Uncaria tomentosa* (300 mg/day) improved the neutropenia caused by chemotherapy, increased superoxide dismutase activity, and restored cellular DNA damage. Therefore, this plant material can be used as a good adjuvant therapy to reduce the adverse effects of chemotherapy for breast cancer patients [144].

There are ongoing clinical trials on the anticancer effects of berberine. A randomized phase II and III trial of berberine hydrochloride to prevent colorectal adenomas in patients with previous colorectal carcinoma is ongoing in China [145]. Using berberine chloride in preventing colorectal cancer in patients with ulcerative colitis in remission, a phase I clinical trial was conducted by National Cancer Institute (NCI) [146]. No results have been published from these two studies. A non-randomized controlled trial on seven familial adenomatous polyposis patients who underwent berberine treatment (300 mg, three times per day for 6 months) showed a significant reduction in the formation and recurrence of polyps in these patients [147]. Meisoindigo, in phase II clinical trial, was used to treat chronic myelogenous leukemia (CML). 134 CML patients were treated with this compound at a dose of 75–150 mg/d. The data evidenced the complete hematological response (CR) and partial response (PR) rates of 32.1% and 48.5%, respectively [148]. Moreover, a phase III clinical study of the effect of meisoindigo at a dose of 100–150 mg/d on the treatment of CML patients showed the hematological CR and PR rates of 45.0% and 39.3% for newly diagnosed patients and 35.9% and 41.4% for pretreated patients [149].

Using green tea catechins (GTCs) in human volunteers with high-grade prostate intraepithelial neoplasia (HG-PIN), a study demonstrated that daily treatment of patients with three GTCs capsules, 200 mg each (total 600 mg/day) for one year, markedly decreased the incidence of the tumor (up to 3%), international prostate symptom score, and quality of life scores. However, prostate-specific antigen (PSA) levels had no significant difference between treated and control patients [150]. Results of another double-blind placebo-controlled phase II clinical study in Italy on the effects of GTCs (600 mg/day) on 60 patients with HG-PIN for one year revealed a significant reduction in the level of PSA in the GTCs-treated group. However, there was no statistically significant difference in disease incidence, improvement in lower urinary tract symptoms, and quality of life between the two arms of the study [151]. Epigallocatechin gallate (EGCG) is the major polyphenol compound in green tea with the ability to prevent the reoccurrence of polyps after polypectomy in colorectal cancer patients. Stingl et al. designed a randomized, placebo-controlled, phase II clinical study to examine the effect of 300 mg/day EGCG on the recurrence of colon adenomas in Germany. They enrolled 1001 patients who had undergone polypectomy for colonic polyps to receive either 150 mg EGCG twice a day or a placebo over the course of 3 years. The study was completed in July 2019, but to date, no study results have been added to clinicaltrials.gov for the study [152].

Clinical testing of shikonin for 19 patients with later-stage lung cancer discovered that the compound markedly decreased the growth of the tumor (to 25% in diameter) and improved the immune system of the patients. The survival rate of one year was 47.3%, effective rate 63.3%, and remission rate 36.9% [153]. In phase I clinical trial to study the multiple-dose safety and anticancer effects of ursolic acid liposomes (UAL) treatment on subjects with advanced solid tumors, some promising outcomes were obtained. Treatment of the patients with various doses of UAL (56, 74, and 98 mg/m^2^) was performed for 14 consecutive days of a 21-day treatment cycle. Although no CR or PR occurred, about 60% of subjects with advanced solid tumors had stable disease. Furthermore, UAL at a dose of 98 mg/m^2^ significantly improved the disease condition and decreased the lesion size in one lung cancer patient [154]. Figure 4 depicts the chemical structure of some major examples of phytochemicals with proapoptotic potential.

### 2.6. Concluding Remarks

The vast majority of experimental data in the literature point out the proapoptotic effects of plant-derived natural products, both the crude extracts and isolated phytochemicals, on human cancer cells through specific targeting of the intrinsic pathway of apoptosis [28]. Targeting a single pathway in battling against cancer may lead to the failure of drugs if this mechanism is interrupted or altered due to various cancer-related phenomenons. Therefore, this can be a drawback for the mentioned approach of cancer therapy and may easily confer resistance to cancer therapeutics. Moreover, a variety of immune system components depends on apoptosis, and hence the changes in its signaling pathways render a resistance phenotype to the immune system. Probable defects in some role-players of apoptosis signaling pathways also raise the threshold for chemotherapy or radiotherapy and lead to resistance to any therapeutic agent [155]. Thus, seeking the novel and more effective natural anticancer therapeutics that simultaneously target both the intrinsic and extrinsic pathways of apoptosis and also block other cross-talks between these pathways and others, such as plant extracts and molecules described in this review, can be a promising and efficient way to combat cancer.

## Figures and Tables

**Figure 1 biomolecules-11-00534-f001:**
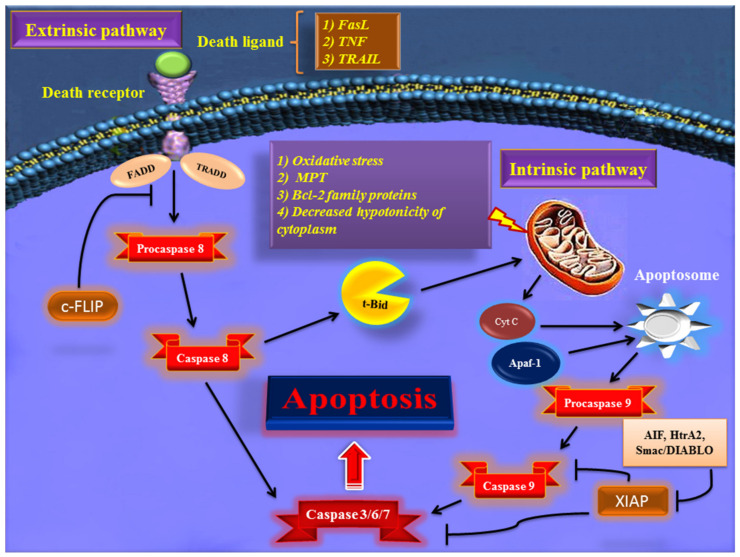
Overview of the extrinsic and intrinsic apoptosis signaling pathways and their major role-players. FasL—Fas ligand; TNF—tumor necrosis factor; TRAIL—TNF- related apoptosis-inducing ligand; TRADD—TNF receptor-associated death domain; FADD—Fas-associated death domain; Bcl-2—B cell lymphoma-2; Cyt c—cytochrome c; Apaf-1—apoptotic protease activating factor 1; AIF—apoptosis-inducing factor; Smac/DIABLO—second mitochondria-derived activator of caspase/direct IAP binding protein with low pI; HtrA2—Omi/high-temperature requirement protein A; XIAP—X-linked inhibitor of apoptosis protein; c-FLIP—cellular FADD-like IL-1β-converting enzyme (FLICE) inhibitory protein; MPT—mitochondrial permeability transition; *t*-BID—truncated BH3 interacting-domain death agonist.

**Figure 2 biomolecules-11-00534-f002:**
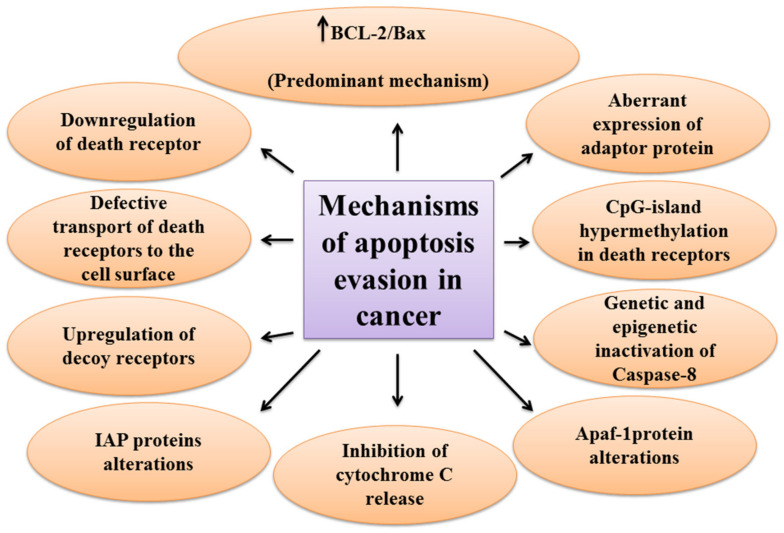
A schematic representation of the different mechanisms of apoptosis evasion used by cancers.

**Figure 3 biomolecules-11-00534-f003:**
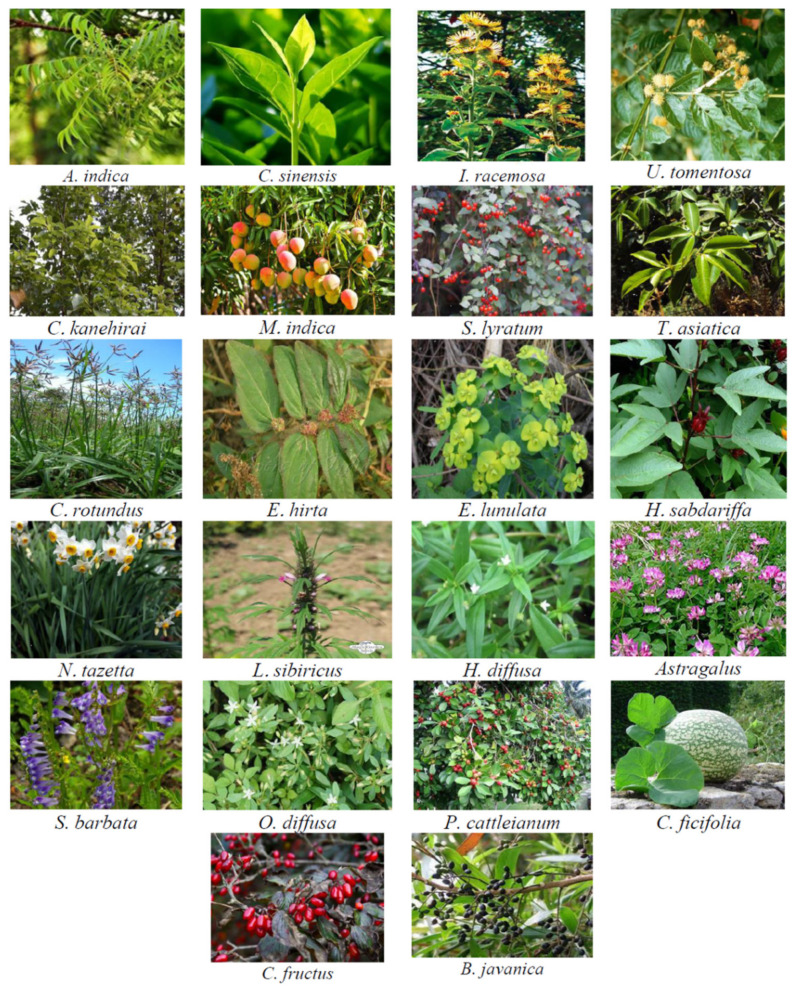
The picture shows representative plants with known proapoptotic activities.

**Figure 4 biomolecules-11-00534-f004:**
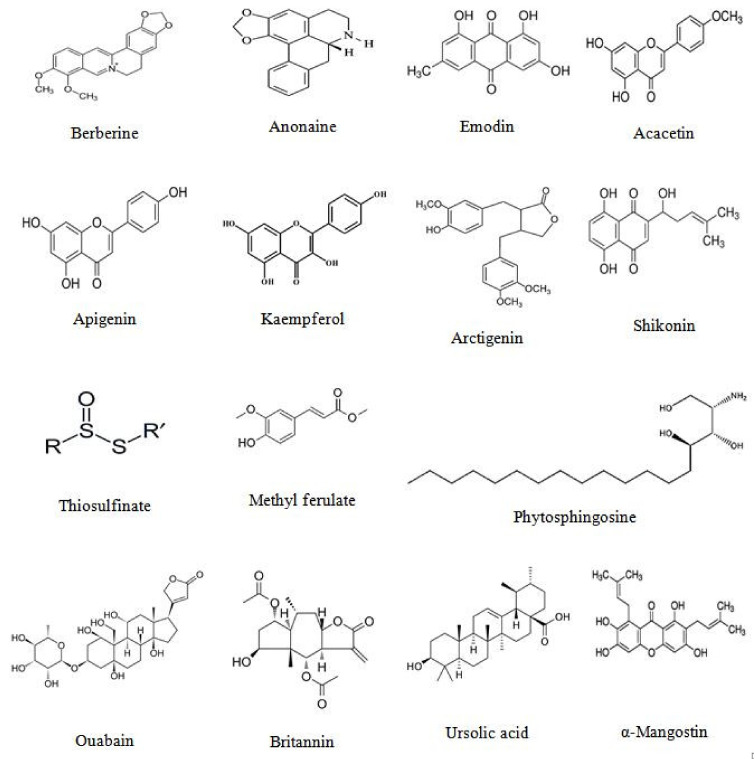
The chemical structure of some major examples of phytochemicals with proapoptotic potential.

**Table 1 biomolecules-11-00534-t001:** Plant extracts with the capacity to trigger both pathways of apoptosis.

Plant Botanicalname	Extraction Solvent	Plant Part Used	Concentration	Cell lines or Animal Model Used	Altered Factors
*Azadirachta indica*	Ethanol	Leaf	200 mg/kg BW	squamous cell carcinoma in a hamster model	Increased: Bim; activation of caspase-3 and -8 Decreased: Bcl-2
*Brucea javanica*	Ethanol	Fruit	Different concentrations for each assay (25, 50, and 100 µg/mL)	HT29	Increased: Fas; TNFR1; TNF2; DR6; CD40; Bid; caspase-8; caspse-9; TRAIL-4; Bax; Bad; cytochrome c release Decreased: Bcl-2
*Camellia sinensis*	Water	Leaf	IC_50_ = 86.68 ± 0.73 μg/mL	HT-29	Increased: activation of caspase-3, -9, and -8 Decreased: NR
*Camellia sinensis*	Water	Leaf	15 × 10^5^ μg/day	Clinical trial, patients with colorectal cancer	Increased: NR Decreased: incidence of metachronous adenomas; size of relapsed adenomas
*Camellia sinensis*	Water	Leaf	9 × 10^5^ μg/day	Clinical trial, patients with metachronous colorectal adenoma and cancer	Increased: NR Decreased: incidence of metachronous adenomas; the number of relapsed adenomas
*Cinnamomum kanehirai Hayata*	Ethanol	Leaf	Different concentrations for each assay (0.25–1.0 mg/mL)	HepG2 and HA22T/VGH	Increased: activation of caspase-3, -9, and -8; Bax Decreased: Bcl-2
*Corni Fructus*	Water	Whole plant	2500 μg/mL	U-2OS	Increased: Bax; cytochrome c release; AIF, Fas, TRAIL; activity and protein level of caspase-3, -9, and -8 Decreased: MMP
*Cucurbita ficifolia*	chloroform	Fruit	IC_50_ = 90 μg/mL	MCF-7	Increased: FADD; BAK; BAX; caspase-3, -9, and -8 Decreased: NR
*Cucurbita ficifolia*	Ethanol	Seed	32 × 10^4^ μg/day	Clinical trial, patients with symptomatic benign prostatic hyperplasia	Increased: quality of life score; maximal urinary flow rate Decreased: international prostate symptom score; Serum prostate-specific antigen
*Cyperus rotundus L.*	Ethanol	Rhizome	200 μg/mL	MDA-MB-231	Increased: Bax; DR5; activation of Bid; activation of caspase-3, -9, and -8 Decreased: Bcl-2; survivin; MMP
*Euphorbia hirta L.*	Methanol	Whole plant	IC_50_ = 25.26 µg/mL	MCF-7	Increased: activation of caspase-2, -6, -8, -9, and -3 Decreased: NR
*Euphorbia lunulata*	n-hexane	Aerial parts	IC_50_ = 20 μg/mL	SGC7901/ADR	Increased: Bax; activation of caspase-3, -9, and -8; cytochrome c release Decreased: Bcl-2
*Hibiscus sabdariffa*	Water	Leaf	50 and 100 μg/mL	LNCaP and LNCaP xenograft nude mice	Increased: Bax; cytochrome c release; activation of caspase-3, -9, and -8; activation of Bid; FasL Decreased: Bcl-2; MMP
*Hwang-Heuk-San (HHS)*	Water	Polyplant formula	Different concentrations for each assay (0–6.1 mg/mL)	HCT116	Increased: Bax; cytochrome c release; activation of caspase-3, -9, and -8; activation of Bid; FasL; DR4; DR5 Decreased: Bcl-2; MMP
*Inula racemosa Hook.f.*	Ethanol	Root	IC_50_ = 16.70 mg/mL for n-hexane fraction	HL-60	Increased: activation of caspase-3, -9, and -8; cytochrome c release; Bax translocation Decreased: MMP
*Leonurus sibiricus*	Methanol	Root	IC_50_ = 1 mg/mL	Grades (I-III) of human glioma cells derived from patients	Increased: Bax; p53; caspase-3, -8, and -9 Decreased: Bcl-2; MMP
*Mangifera indica*	Ethanol	Fruit peel	Different concentrations for each assay (0–400 µg/mL)	HeLa	Increased: activation of caspase-3, -9, and -8 Decreased: Bcl-2
*Narcissus tazetta var. chinensis*	Chloroform	Stem and leaf	5.0 μg/mL	HL-60	Increased: Bax; cytochrome c release; activation of caspase-3, -9, and -8 Decreased: Bcl-2
*Oldenlandia diffusa*	Methanol and butanol	Whole plant	Different concentrations for each assay (0–20 µg/mL) Oldenlandia diffusa	MCF-7	Increased: Bax; activation of caspase-8 and -7 Decreased: Bcl-2
*Psidium cattleianum Sabine*	Chloroform	Leaf	Different concentrations for each assay (0–200 µg/mL) Oldenlandia diffusa	SNU-16	Increased: Bax; PARP; caspase-3 and -8 Decreased: Bcl-2
*Qingjie Fuzheng granule (QFG)*	Water	Polyplant formula	Different concentrations for each assay (0–1500 µg/mL) for cell lines; 0.75 g/kg and 1.5 g/kg for mice	SK-Hep-1, Bel-7402, HCT-116, and HCT-8; mouse xenograft model	Increased: Fas; FasL; Bax; activation of caspase-3, -9, and -8 Decreased: Bcl-2; tumor weight in mice
*Solanum lyratum*	Chloroform	Whole plant	40 μg/mL	HSC-3, SAS, and CAL-27	Increased: Bax and Bad; activation of caspase-3, -9, and -8 Decreased: Bcl-2 and Bcl-xl; MMP
*So-Cheong-Ryong-Tang*	Water	Polyplant formula	500 and 1000 μg/mL for cell line; 157.5 mg/kg/day for mice	AGS; mouse xenograft model	Increased: activation of caspase-3, -9, and -8 Decreased: tumor weight in mice
*Toddalia asiatica (L.) Lam.*	Dichloromethane	Root	IC_50_ = 18 μg/mL	HT-29	Increased: activation of caspase-3, -9, and -8 Decreased: NR
*Uncaria tomentosa (Wild.) DC.*	Ethyl acetate	Whole plant	100 μg/mL	HL-60	Increased: Fas, activation of caspase-3, -9, and -8; Bax; cytochrome c release Decreased: MMP; Bcl-XL
*Uncaria tomentosa (Wild.) DC.*	Ethanol	Bark	30 × 10^4^ μg/day	Clinical trial, patients with breast cancer	Increased: Neutrophil count; Superoxide dismutase activity Decreased: DNA damage

MMP: Mitochondrial membrane potential; NR; Not reported.

**Table 2 biomolecules-11-00534-t002:** Phytochemicals with the ability to induce both pathways of apoptosis.

Chemical Family	Molecule Name	Concentration (µM)	Cell Line	Altered Factors
Alkaloid	(-)-Anonaine	100 μM	HeLa	Increased: Bax; cytochrome C release; activation of caspase-3, -7, -9, and -8 Decreased: MMP
	Berberine	IC_50_ = 75 μM	SCC-4	Increased: Bax; cytochrome C release; activation of caspase-3, -9, and -8; AIF; Endo G Decreased: MMP Bcl-2
		30 × 10^4^ μg/day	Clinical trial, patients with familial adenomatous polyposis	Increased: NR Decreased: polyp size and number
	Hemanthamine and hemanthidine	Various concentrations (5–20 μM)	Jurkat	Increased: activation of caspase-3, -7, -9, and -8 Decreased: MMP
	Lycorine	IC_50_ = 1 µM	HL-60	Increased: Bax; activation of caspase-3, -9, and -8 Decreased: Bcl-2
		IC_50_ = 1.25 μM	KM3	Increased: Bax; activation of caspase-3, -9, and -8 Decreased: Bcl-2
	Meisoindigo	20 µM	HL-60	Increased: Bax; cytochrome C release; activation of caspase-3, -9, and -8; FasL Decreased: Bcl-2
		75–150 × 10^3^ µg/day	Clinical trial phase II, patients with chronic myelogenous leukemia	Increased: Hematological complete response (CR) and partial response (PR) rates of 32.1% and 48.5%, respectively Decreased: NR
		100–150 × 10^3^ µg/day	Clinical trial phase III, patients with chronic myelogenous leukemia	Increased: hematological CR and PR rates of 45.0% and 39.3% for newly diagnosed patients and 35.9% and 41.4% for pretreated patients Decreased: NR
	6-methoxydihydrosanguinarine	IC_50_ = 3.8 µM	HepG2	Increased: Bax; cytochrome C release; activation of caspase-3, -9, and -8 Decreased: Bcl-2
	Sanguinarine	Various concentrations (0.25–4 μM)	BC1, BC3, BCBL1, and HBL6	Increased: Bax; cytochrome C release; activation of caspase-3, -9, and -8; activation of Bid; DR4 Decreased: MMP
	Tetrandrine and Cepharanthine	Various concentrations (3–15 μM)	Jurkat	Increased: Bax; activation of caspase-3, -6, -9, and -8 Decreased: Bcl-2
Anthraquinone	Emodin	IC_50_ = 9.06 μM for MDA-MB-453 cellsIC_50_ = 0.83 μM for Calu-3 cells	MDA-MB-453 and Calu-3	Increased: cytochrome C release; activation of caspase-3, -9, and -8; activation of Bid Decreased: MMP
		40 mg/kg/once every 3 days	LS1034 colon cancer cells xenografts into male athymic BALB/c nu/nu mice	Increased: NR Decreased: tumor volume
Flavonoid	Acacetin	IC_50_ = 60 µM	AGS	Increased: Bax; cytochrome C release; activation of caspase-3, -9, and -8; activation of Bid and Bad; FasL; Fas Decreased: Bcl-2; MMP
	Ampelopsin	IC_50_ = 39.6 µM for U251IC_50_ = 35.8 µM for A172	U251 and A172	Increased: activation of caspase-3, -9, and -8 Decreased: NR
		50 and 100 mg/kg/day for 30 days	U251 bearing BALB/c-nu mice	Increased: activation of caspase-3, -9, and -8; PARP Decreased: tumor volume and progression
	Anthocyanins	Various concentrations 0–265.4 µM	U937	Increased: Bax; activation of caspase-3, -9, and -8; activation of Bid Decreased: Bcl-2; XIAP; cIAP-1; cIAP-2; MMP
	Apigenin in combination with TRAIL	IC_50_ = 20 μM	A549 and H1299	Increased: Bax; Bad; DR4; DR5 Decreased: Bcl-2; Bcl-xL
		10 μg/mouse	Tumor xenografts A549	Increased: DR4; DR5; apoptotic and necrotic cell death Decreased: tumor volume
	Casticin	IC_50_ = 0.85 µM	HT-29, HCT-116, and SW480	Increased: Bax; activation of caspase-3; DR5; activation of Bid Decreased: Bcl-2; Bcl-xL; XIAP; cFLIP
	Catechins in green tea	6 × 10^5^ μg/day	Clinical trial, patients with high-grade prostate intraepithelial neoplasia	Increased: NR Decreased: incidence of the tumor, international prostate symptom score, and quality of life scores
	Catechins in green tea	6 × 10^5^ μg/day	Clinical trial, patients with high-grade prostate intraepithelial neoplasia	Increased: NR Decreased: prostate-specific antigen (PSA)
	Epigallocatechin gallate	3 × 10^5^ μg/day	Clinical trial, patients with metachronous colon adenomas	Increased: NR Decreased: NR
	Eupafolin	IC_50_ = 26.75 μM	HeLa	Increased: cytochrome C release; activation of caspase-3, -6, -7, -9, and -8 Decreased: Bcl-2; MMP
	Fisetin	Various concentrations (0–100 μM)	MCF-7	Increased: activation of caspase-7, -9, and -8 Decreased: MMP
		Various concentrations (0–100 μM)	MDA-MB-468 and MDA-MB-231	Increased: activation of caspase -9 and -8 Decreased: NR
		Various concentrations (10–60 μM)	LNCaP	Increased: cytochrome C release; activation of caspase-3, -9, and -8 Decreased: Bcl-2; XIAP
		223 mg/kg/day for two weeks	LLC bearing C57BL/6 J female mice	Increased: NR Decreased: tumor volume and angiogenesis
	Isoangustone A	Various concentrations (2.4–17.7 μM)	DU145	Increased: cytochrome C release; activation of caspase-3, -7, -9, and -8; activation of Bid; Fas; DR4 Decreased: MMP
	Kaempferol	Various concentrations (20–100 μM)	OVCAR-3 and SKOV-3	Increased: Bax; activation of caspase-3, -9, and -8 Decreased: Bcl-2; Bcl-xL; XIAP; cFLIP
	Morusin	IC_50_ = 6.1 µM	HT-29	Increased: Smac/DIABLO; cytochrome C release; activation of caspase-3, -9, and -8 Decreased: XIAP; MMP
	Wogonin	IC_50_ = 75 µM	U-2OS	Increased: Bax; Bad; cytochrome C release; activation of caspase-3, -4, -9, and -8; AIF; Endo G; Fas Decreased: NR
Lignin	Arctigenin	IC_50_ = 0.24 μM	Hep G2 and SMMC7721	Increased: Bax; cytochrome C release; activation of caspase-3, -9, and -8; FasL; Fas Decreased: Bcl-2; MMP
Naphthoquinone	Plumbagin	IC_50_ = 9 μM	NB4	Increased: Bax; Bak; activation of caspase-3, -9, and -8 Decreased: Bcl-xL; MMP
		2 mg/kg	NB4 cell bearing male NOD/SCID mice	Increased: NR Decreased: tumor volume
	Shikonin	IC_50_ = 4 µM for Huh7 IC50 = 5.3 µM for BEL7402	BEL7402 and Huh7	Increased: activation of caspase-9 and -8; activation of Bid Decreased: Bcl-2; c-FLIP
		5 or 10 mg/kg for 30 days	Huh7 cell bearing male BALB/c nude mice	Increased: activation of caspase-9 and -8, and PARP Decreased: tumor volume
		IC_50_ = 32.5 μM	Tca-8113	Increased: activation of caspase-3, -9, and -8 Decreased: Bcl-2
		5–10 (mg/kg/day)	Clinical trial, patients with later-stage lung cancer	Increased: immune system; survival rate Decreased: tumor growth; remission rate
Organosulfur derivative	Thiosulfinates	IC_50_ = 10.07 μM	PC-3	Increased: Bax; AIF; activation of caspase-3, -9, and -8; activation of Bid; Decreased: Bcl-2
		40 and 80 μM	HT-29	Increased: Bax; AIF; activation of caspase-3, -9, and -8; activation of Bid; Decreased: Bcl-2
Eugenol ortho dimer	Biseugenol B	IC_50_ = 4 μM	PC3	Increased: Bax; cytochrome C release; activation of caspase-3, -7, -9, and -8 Decreased: Bcl-2; MMP
Hydroxycinnamic acids derivative	Methyl ferulate	IC_50_ = 1.73–1.9 μM	SW1116 and SW837	Increased: Bax; Bad; Apaf1; Bid; Bim; Smac; caspase-2, -3, -6, -7, -8, and -9 Decreased: Bcl-2; c-IAP-1; c-IAP-2; FLIP
Phospholipid	N, N-dimethyl Phytosphingosine	Various concentrations (0–7.5 μM)	HL-60	Increased: activation of caspase-3, -9, and -8; cytochrome C release Decreased: Bcl-2; MMP
	Phytosphingosine	15.8 or 31.5 μM	Jurkat and NCI-H460	Increased: Bax translocation to mitochondria; cytochrome C release; activation of caspase-3, -9, and -8 Decreased: MMP
Steroid	Oleandrin	Various concentrations (0–0.05 μM)	U-2OS and SaOS-2	Increased: Bax; cytochrome C release; activation of caspase-3, -9, and -8; FasL; Fas Decreased: Bcl-2; MMP
	Ouabain	IC_50_ = 5 μM	U-2OS	Increased: Bax; cytochrome C release; activation of caspase-3, -9, and -8; AIF; Endo G Decreased: Bcl-2; MMP
		2 mg/kg/day for 13 days	Mouse model of xenografted SH-SY5Yneuroblastoma cells	Increased: activation of caspase-3 Decreased: tumor volume
Terpene	Britannin	Various concentrations (0–80 μM)	SMMC-7721 and HepG2	Increased: activation of caspase-3, -9, and -8 Decreased: Bcl-2
		Various concentrations (0–30 mg/kg/day for 21 days)	HepG2 bearing male BALB/c nu/nu nude mice	Increased: p-AMPK, cleaved caspase-3 and LC3 II Decreased: p-mTOR; Ki-67; tumor volume
	Celastrol	IC_50_ = 2.12 μM	A549	Increased: Bax; cytochrome C release; activation of caspase-3, -9, and -8; FasL; Fas Decreased: Bcl-2
		IC_50_ = 2.55 μM for HOS IC_50_ = 1.97 μM for MG-63	HOS and MG-63	Increased: activation of caspase-3, -9, and -8; activation of Bid; DR5 Decreased: MMP
		4.5 mg/kg/day for 28 days	Xenografts of glioma SHG44 cells in female BALB/c mice	Increased: NR Decreased: tumor growth
	Corosolic acid	IC_50_ = 28 μM	HeLa	Increased: Bax; cytochrome C release; activation of caspase-3, -9, and -8 Decreased: Bcl-2; MMP
	Dehydrocostus lactone	8.7 μM	DU145	Increased: Bax; Bak; Bok; Bik; Bmf; *t*-Bid; activation of caspase-3, -9, and -8 Decreased: Bcl-xL
	Galbanic acid in combination with TRAIL	Various concentrations (0–50 μM)	H460/R	Increased: activation of caspase-9 and -8; DR5; activation of Bid Decreased: Bcl-2; Bcl-xL; XIAP
	Lambertianic acid in combination with TRAIL	IC_50_ = 20 μM	A549 and H1299	Increased: activation of caspase-3, -9, and -8; DR4; activation of Bid Decreased: Bcl-2; XIAP; cFLIP
	Myriadenolide	IC_50_ = 30 μM	Jurkat and THP-1	Increased: activation of caspase-3, -9, and -8; activation of Bid Decreased: MMP
	Nimbolide	IC_50_ = 5 µM	DU-145, PC-3, A-549	Increased: activation of caspase-3, -9, and -8 Decreased: NR
	Raddeanin A	IC_50_ = 5.34 µM for BGC-823, IC_50_ = 6.61 µM for SGC-7901, and IC_50_ = 4.98 μM for MKN-28	BGC-823, SGC-7901, and MKN-28	Increased: Bax; activation of caspase-3, -9, and -8 Decreased: Bcl-2; Bcl-xL
		Different concentrations of raddeanin A (0.5, 1.5, and 4.5 mg/kg)	Granuloma cell line S180, hepatic carcinoma cell line H22, and cervical cancer cell line U14 mice models	Increased: NR Decreased: tumor volume of granuloma cell line S180, hepatic carcinoma cell line H22, and cervical cancer cell line U14 models
	Rosamultic acid	Various concentrations (0–100 μM)	SGC-7901	Increased: activation of caspase-3, -9, and -8 Decreased: NR
	Saikosaponin A	IC_50_ = 20 μM	LoVo, SW48	Increased: Bax; activation of caspase-3, -2, -9, and -8; activation of Bid Decreased: Bcl-2; MMP
	Saponins	30.3 μM	HT-29	Increased: Bax; activation of caspase-3, -9, and -8; activation of BidDecreased: Bcl-2
	Tubeimoside-1	Various concentrations (0–40 μM)	HepG2	Increased: Bak; activation of caspase-3, -9, and -8; Fas; FasL Decreased: Bcl-2; MMP
	Ursolic acid	40 μM	RC-58 T/h/SA#4	Increased: Bax; activation of caspase-3, -9, and -8; activation of Bid Decreased: Bcl-2
		56, 74, and 98 mg/m^2^	Clinical trial; patients with advanced solid tumors	Increased: 60% of patients had stable disease; 1 lung cancer patient showed significant improvement Decreased: The lesion size
Xanthone	α-Mangostin	IC_50_ = 24.9 µM	MCF7	Increased: Bax; cytochrome C release; activation of caspase-3, -7, -9, and -8 Decreased: Bcl-2; MMP
		30 and 60 mg/kg	LA7 cells bearing female Sprague-Dawley rats	Increased: NR Decreased: tumor volume
	Pyranocycloartobiloxanthone A	IC_50_ = 1.4 µM	MCF7	Increased: Bax; cytochrome C release; activation of caspase-3, -7, -9, and -8 Decreased: Bcl-2; MMP

MMP—mitochondrial membrane potential; NR—not reported.

## Data Availability

Not applicable.

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
