# Peer review of "The Most Competent Plant-Derived Natural Products for Targeting Apoptosis in Cancer Therapy"

_biomolecules, 2021, doi:10.3390/biom11040534_

Round 1

Reviewer 1 Report

The article titled “The most competent plant-derived natural products to target apoptosis in cancer therapy” offers a review on natural products' application for cancer treatment as well as their mechanisms of action. The topic was well chosen, as it is of growing interest to the scientific community. The article is well written, easy to read, and with the right amount of introduction of apoptotic and evasion pathways to further approach the primary topic.

In order to improve the manuscript, the authors should address some critical points:

  1. English language revision and spell check are recommended (e.g. “in vitro” in italic, “bleb” expression, and minor typos along with the text)
  2. The authors provide a section for both extrinsic and intrinsic pathways of apoptosis, approaching molecular targets involved in the process that are not mentioned in figure 1. To benefit the reader, figure 1 could be improved in both image quality and to include more steps of the metabolic pathways depicted.
  3. Section 2.3 could benefit from the addition of a figure.
  4. Please revise the sentence “In a clinical trial, treating the patients with previous colorectal adenomas and confirmed clean colon a year after removal of the adenomas with green tea extract showed a reduction in the incidence of metachronous adenomas and the size of relapsed adenomas in comparison to control group [61].” (page 5)
  5. The authors carefully added common names to the species mentioned in the manuscript. In my opinion, the expression “(known as there is no common name)” is not needed often in the text, as it could be referred at the first appearance that species without the information do not have a common name.
  6. In table 2, the compounds are listed by class. The authors should revise the order, as flavonoids are also phenolic compounds. A different nomenclature could be used.
  7. Regarding natural products already used in human clinical trials or presently in clinical trials could be rearranged in a new section, providing a separation from studies obtained from in vitro sources, in vivo, and human clinical trials. This new information would improve the manuscript as it would describe natural compounds within all the phases of testing for the development of cancer treatment drugs.

Author Response

We sincerely thank the reviewer for constructive criticisms and valuable comments, which were of great help in revising the manuscript. Accordingly, the revised manuscript has been systematically improved with new information and additional corrections. Our responses (AC) to the referee’s comments (RC) are given below.

Reviewer 1

The article titled “The most competent plant-derived natural products to target apoptosis in cancer therapy” offers a review on natural products' application for cancer treatment as well as their mechanisms of action. The topic was well chosen, as it is of growing interest to the scientific community. The article is well written, easy to read, and with the right amount of introduction of apoptotic and evasion pathways to further approach the primary topic.

In order to improve the manuscript, the authors should address some critical points:

(RC): English language revision and spell check are recommended (e.g. “in vitro” in italic, “bleb” expression, and minor typos along with the text)

(AC): Thank you very much for pointing out theses error. We have revised the whole manuscript for any grammatical and typographical mistakes and highlighted the corrections in yellow color.

(RC): The authors provide a section for both extrinsic and intrinsic pathways of apoptosis, approaching molecular targets involved in the process that are not mentioned in figure 1. To benefit the reader, figure 1 could be improved in both image quality and to include more steps of the metabolic pathways depicted.

(AC): According to your suggestion, we have added other molecules involved in the apoptosis signaling pathway.

(RC): Section 2.3 could benefit from the addition of a figure.

(AC): According to your suggestion, we have designed another figure for section 2.3.

 (RC): Please revise the sentence “In a clinical trial, treating the patients with previous colorectal adenomas and confirmed clean colon a year after removal of the adenomas with green tea extract showed a reduction in the incidence of metachronous adenomas and the size of relapsed adenomas in comparison to control group [61].” (page 5)

(AC): In accordance with your suggestion, we have changed this sentence and highlighted it in yellow.

(RC): The authors carefully added common names to the species mentioned in the manuscript. In my opinion, the expression “(known as there is no common name)” is not needed often in the text, as it could be referred at the first appearance that species without the information do not have a common name.

(AC): We appreciate this observation. I have removed these errors in the manuscript.

(RC): In table 2, the compounds are listed by class. The authors should revise the order, as flavonoids are also phenolic compounds. A different nomenclature could be used.

(AC): Thank you very much for your suggestion. We have changed the chemical name of both Biseugenol B and Methyl ferulate to eugenol ortho dimer and Hydroxycinnamic acids derivative, respectively.

(RC): Regarding natural products already used in human clinical trials or presently in clinical trials could be rearranged in a new section, providing a separation from studies obtained from in vitro sources, in vivo, and human clinical trials. This new information would improve the manuscript as it would describe natural compounds within all the phases of testing for the development of cancer treatment drugs.

(AC): In accordance with your excellent suggestion, we have added a new section as “Use of plant extracts and plant molecules in human clinical trials” and highlighted it in yellow.

Reviewer 2 Report

The work entitled “The most competent plant-derived natural products to target apoptosis in cancer therapy” by Rajabi et al. reviews the ability of different extracts and plant-derived molecules to activate both intrinsic and extrinsic apoptosis pathways in human cancer cells, and their potential for cancer treatments.

The subject is well introduced. The authors did a good job exposing the problematics and identifying the targeted apoptosis pathways. The collection of works contemplated in this review about the influence of the extracts on the cancer cells is significant, and detailed. Further, the authors were smart in the presentation of the information since they were able to offer some discussion and a train of though could be formed through it. Overall, the work is detailed and well put together. Some of the works reviewed are a bit old and thus, and as such it would benefit from the addition of more recent work, particularly considering that in the last two years much research has been conducted on the use of natural extracts for biomedical applications, including cancer treatments.

The only minor observations are:

  • Table 1 would benefit from a little more information on the extracts’ activity on the altered factors. The authors should add another section named “Observations” and explore more about their mechanisms of action.
  • Addition of more 2020 research.
  • English writing should be improved.

Author Response

We sincerely thank the reviewer for constructive criticisms and valuable comments, which were of great help in revising the manuscript. Accordingly, the revised manuscript has been systematically improved with new information and additional corrections. Our responses (AC) to the referee’s comments (RC) are given below.

Reviewer 2

The work entitled “The most competent plant-derived natural products to target apoptosis in cancer therapy” by Rajabi et al. reviews the ability of different extracts and plant-derived molecules to activate both intrinsic and extrinsic apoptosis pathways in human cancer cells, and their potential for cancer treatments. The subject is well introduced. The authors did a good job exposing the problematics and identifying the targeted apoptosis pathways. The collection of works contemplated in this review about the influence of the extracts on the cancer cells is significant, and detailed. Further, the authors were smart in the presentation of the information since they were able to offer some discussion and a train of though could be formed through it. Overall, the work is detailed and well put together. Some of the works reviewed are a bit old and thus, and as such it would benefit from the addition of more recent work, particularly considering that in the last two years much research has been conducted on the use of natural extracts for biomedical applications, including cancer treatments.

The only minor observations are:

(RC): Table 1 would benefit from a little more information on the extracts’ activity on the altered factors. The authors should add another section named “Observations” and explore more about their mechanisms of action.

(AC): With all due respect to the reviewer, we would like to explain why we do not agree with this suggestion. As you see in the altered factors section in this table, the mechanisms of proapoptotic activities of all extracts are included in the mentioned section. Indeed, this section in the mentioned table clearly shows the exact mechanism of extracts to induce apoptosis in cancer cells. Therefore, we believe that there does not appear to explore any mechanism for proapoptotic effects of the extracts.

(RC): Addition of more 2020 research.

(AC): We appreciate the reviewer’s perspective, but we should say that we have searched and gathered all of studies, which show the effects of plant-derived products on both pathways of apoptosis. So, we would like to explain that this present review article completely covers all relevant studies published in different years.

(RC): English writing should be improved.

(AC): Thank you very much for pointing out this error. We have revised the whole manuscript for any grammatical and typographical mistakes and highlighted the corrections in yellow color.

Regards

Dr Marc MARESCA

Reviewer 3 Report

I personally like the effort of the authors trying to collect a huge amount of information and put it together in a concise and summarize way.

Please find our comments in the attached document. 

Author Response

We sincerely thank the reviewer for constructive criticisms and valuable comments, which were of great help in revising the manuscript. Accordingly, the revised manuscript has been systematically improved with new information and additional corrections. Our responses (AC) to the referee’s comments (RC) are given below.

Reviewer 3

I personally like the effort of the authors trying to collect a huge amount of information and put it together in a concise and summarize way. Plant-derived natural products, in especial secondary metabolites, are one of the most promising alternatives to the traditional therapies against tumoral pathologies. In general, the manuscript is clear and fluid and I cannot see many discrepancies. However, I have detected some sentences with difficult reading and comprehension. Moreover, in my opinion, there are some sentences with informal language. I would recommend the authors to revise all the manuscript, to solve them. In the following notes; I will provide clear information to improve the manuscript.

 (RC): In terms of information, the authors made a great work explaining the different apoptosis ways and they give a very complete list of different plants and compounds with relevant cytotoxic activity in cancer cells. One problem, in my opinion, it’s that in sections 2.4 the information it´s given as a list of articles without highlighting any of them or the most important aspects. I will suggest some modifications or corrections in order to improve the document.

(AC): With all due respect to the reviewer’s point of view, we would like to explain why we didn’t highlight other aspects of the reviewed articles in this section. Looking at the title of the manuscript and the main goals of paper in the end of introduction section illustrates that our purpose of writing this paper was to review the articles which contain some results associated with simultaneous targeting of both pathways of apoptosis by plant natural products. We agree that additional data and surveys would provide useful and important perspectives for researchers, but we believe that the recommended changes are outside the scope of this article and having an exhaustive review of those aspects may prevent the main goal of this review article to be conveyed to the reader. However, we can incorporate the reviewer’s suggestion into future work.

(RC): I think that this sentence is difficult to read. Please, rewrite the sentence.  “At the later step of apoptosis, the cell shrinks, the cell membrane begins to bleb, the cytoplasmic organelles are modified, and finally the membrane losses its integrity and converted to apoptotic bodies”

(AC): Thank you for this excellent suggestion. We have revised the sentence and highlighted it in yellow color.

(RC): The expression “eat me” describes very well the process that is carried on in the early stage of apoptosis when the phosphatidylserine is situated on the outside of the cell membrane. Nevertheless, in my opinion, it´s inappropriate in an explication of these characteristics. Please, if it is possible try to change the expression “eat me”.

(AC): In accordance with the reviewer’s suggestion, we have changed this phrase to “engulfment signal” and highlighted it in yellow color.

(RC): In the sentences “…push cancer to escape apoptosis…” and “… to scape apoptotic cell death…”, in my opinion, the verbs push and scape are not well used in this context, please, use other verbs as inhibit.

(AC): In accordance with the reviewer’s suggestion, we have changed these two sentences to “compel cancers to avoid apoptosis” and “to evade apoptosis”, respectively, and highlighted it in yellow color.

(RC): Correct me if I am wrong but, in this case, it should not be upregulated instead of downregulated.

(AC): Thank you very much for pointing out this error. We have corrected it in the text and highlighted in yellow.

(RC): This sentence it´s difficult to read. Please rewrite it.” This product acted by elevating the activity of caspase-9, -3, and -8 together with the decreased mitochondrial membrane potential (MMP), cytochrome C release, Bax translocation to the mitochondrial membrane [62].”

(AC): In accordance with your comment, we have revised this sentence to improve it and highlighted it in yellow color.

 (RC): In this table, since the plant Azadirachta indica until the Hibiscus sabdariffa, is so difficult to differentiate the altered factors of every plant. Please, try to separate the information as you did since the plant Hwang-Heuk-San.

(AC): Thank you very much for your suggestion. We have corrected them in the table.

(RC): There is a format error.

(AC): Thank you for pointing out the error. We have corrected it and highlighted in yellow.

(RC): The gastric cell lines used are cancer gastric cell lines. Please, write it.

(AC):  Thank you very much for pointing out the error. We have corrected it and highlighted in yellow.

(RC): This should be in parentheses. (NB4)

(AC): In accordance with your suggestion, we have put it in the parentheses and highlighted in yellow.

(RC): In my opinion, use the expression “Achilles heel” it´s not so appropriate. Please try to find another way to say it.

(AC): In accordance with your suggestion, we have replaced it by “drawback” and highlighted it in yellow.

Regards

Dr Marc MARESCA

Round 2

Reviewer 3 Report

Review V2 Biomolecules-1137259

Title: The most competent plant-derived natural products to target apoptosis in cancer therapy

I appreciate the suggested changes the authors have incorporated to the revised version of the manuscript as I believe it has made it more comprehensive and enhanced its quality. Moreover, fig. 2 and section 2.5 increase in a very effective and concrete way relevant information on current advances in cancer treatment with plant-derived natural products. However, I have some new suggestions in the new section 2.5.

In this sentence “Therefore, this plant material can be used as a good adjuvant therapy to reduce the adverse effects of the chemotherapy for breast cancer patients [144]” the article “the” could be removed.

In the next sentences the article “a” before “phase” could be removed. “...colitis in remission, a phase I clinical trial was conducted…”, “Meisoindigo, in a phase II clinical trial, was used to treat chronic myelogenous leukemia (CML)”